# Mass spectrometric based detection of protein nucleotidylation in the RNA polymerase of SARS-CoV-2

Brian J. Conti[1,2], Andrew S. Leicht [1,3], Robert N. Kirchdoerfer [1,3] & Michael R. Sussman [1,2✉]

Coronaviruses, like severe acute respiratory syndrome coronavirus 2 (SARS-CoV-2), encode a nucleotidyl transferase in the N-terminal (NiRAN) domain of the nonstructural protein (nsp) 12 protein within the RNA dependent RNA polymerase. Here we show the detection of guanosine monophosphate (GMP) and uridine monophosphate-modified amino acids in nidovirus proteins using heavy isotope-assisted mass spectrometry (MS) and MS/MS peptide sequencing. We identified lysine-143 in the equine arteritis virus (EAV) protein, nsp7, as a primary site of in vitro GMP attachment via a phosphoramide bond. In SARS-CoV-2 replicase proteins, we demonstrate nsp12-mediated nucleotidylation of nsp7 lysine-2. Our results demonstrate new strategies for detecting GMP-peptide linkages that can be adapted for higher throughput screening using mass spectrometric technologies. These data are expected to be important for a rapid and timely characterization of a new enzymatic activity in SARS-CoV-2 that may be an attractive drug target aimed at limiting viral replication in infected patients.

[1] Department of Biochemistry, University of Wisconsin-Madison, Madison, WI, USA. [2] Center for Genomic Science Innovation, University of Wisconsin-Madison, Madison, WI, USA. [3] Institute for Molecular Virology, University of Wisconsin-Madison, Madison, WI, USA. ✉email: msussman@wisc.edu

The world is faced with a pandemic disease, Coronavirus disease 2019 (COVID-19), caused by the emergence and global spread of a new species of coronavirus. Although it is clear that lethality of the disease progression is often caused by pulmonary problems associated with severe respiratory symptoms, this virus continues to surprise the medical community with new pathologies that are creating challenges for effective treatment. While the causative agent of COVID-19, severe acute respiratory syndrome coronavirus 2 (SARS-CoV-2), is similar to other coronaviruses that infect humans and animals, many molecular details of this virus's macromolecular structures and functions remain unknown. Immunological approaches to neutralize the virus, using both passive (e.g., injection of antibodies into patients) or active (e.g., injection of DNA, messenger RNA or protein that generates a neutralizing immune response) immunity are being pursued by scientists in both the private and public sectors. Though it appears successful vaccines have recently been created[1], it is prudent to explore other approaches to treat this disease, including drug therapy. Small molecule drug therapy has been highly successful in controlling human immunodeficiency virus 1 infections and curing infections of hepatitis C virus[2–5]. Creating small molecule drugs that target the SARS-CoV-2 replication machinery would provide a complementary approach to ongoing vaccine development efforts as well as preparing the world for future outbreaks caused by emerging coronaviruses.

A logical target for drugs that inhibit the virus is the conserved protein machinery responsible for replication of the viral RNA genome. Targeting proteins associated with the viral RNA-dependent RNA Polymerase (RdRP) is particularly attractive given the absence of similar RNA replication machinery in human cells. The small molecule Remdesivir targets the coronavirus replication machinery causing premature RNA chain termination[6]. Remdesivir has been approved for the treatment of COVID-19 patients requiring hospitalization[7]. Discovery of new enzymatic and protein binding activities in SARS-CoV-2 is a critical part in elucidating the viral life cycle and importantly, in developing novel strategies to combat this disease as well as potentially emerging viruses that use similar enzymes.

A novel nucleotidylation activity was discovered in the RdRP of equine arteritis virus (EAV) nonstructural protein (nsp) 9 within the newly defined nidovirus RdRP-associated nucleotidyl-transferase (NiRAN) domain that is also conserved in SARS-CoV-2[8–10]. Mutation of residues that were critical in EAV nsp9 nucleotidyl transferase activity, as well as mutation of the corresponding residues in SARS-CoV nsp12, eliminated viral propagation[10]. A recent report proposed that the nsp12 NiRAN domain mediates RNA capping[11], which traditionally involves the transfer of guanosine monophosphate (GMP) to a protein sidechain before its movement to the 5' RNA terminus[12–14]. However, there has been no definitive direct chemical analysis of the bond covalently linking the nucleotides to the protein, the exact chemical structure of the modification or the amino acids in which it occurs. Here, we characterize the nucleotidyl transferase activity in SARS-CoV-2 replicase proteins by detecting GMP-protein and uridine monophosphate (UMP)-protein adducts using mass spectrometric technologies. The SARS-CoV-2 RdRP nsp12 mediated the nucleotidylation of nsp7 and nsp8, which bind near the nsp12 RNA polymerase active site and act as essential co-factors in RNA synthesis[9,15].

## Results

### SARS-CoV-2 nsp12 and EAV nsp9 mediate GMP protein labeling.
In order to utilize mass spectrometric technologies to provide definitive chemical data on the nucleotidylation event in nidoviruses, we first confirmed the earlier report of nucleotidyl transferase activity in EAV. We incubated EAV nsp9 (relative molecular mass (Mr) ~76,800) alone or together with α-$^{32}$P-guanosine triphosphate (GTP) and a 2–4 molar excess of nsp7 (Mr~25,200), which binds nsp9 and plays an important role in viral propagation (Fig. 1a, Supplementary Fig. 1)[16,17]. Nucleotidyated products were analyzed by sodium dodecyl sulfate–polyacrylamide gel electrophoresis (SDS-PAGE) followed by Coomassie staining and autoradiography. The same analysis was performed on SARS-CoV-2 nsp7 (Mr~9,300), nsp8 (Mr~21,900) and nsp12 (Mr~106,700) that represent the minimal core protein subunits comprising the RdRP (Fig. 1b, Supplementary Fig. 1) (note that by convention, the numbering of nonstructural proteins that comprise the RdRP of EAV is independent of that in SARS-CoV-2 and reflects their position within the viral polyprotein, not their function). The proteins were kept at low temperatures and neutral pH to stabilize labile, phosphoramide linkages between GMP and protein (Supplementary Fig. 2)[10,14,18]. Although EAV nsp9 alone was sufficient to observe protein nucleotidylation[10], the addition of nsp7 resulted in higher levels of radionucleotide incorporation into both proteins. In contrast, full length SARS-CoV-2 nsp12 never incorporated radioactivity itself but was necessary to observe nucleotidylation on nsp7 and nsp8. In EAV nsp9, residue R124 is critical for nucleotidyl transferase activity[8,19]. Mutation of the homologous R116 residue in SARS-CoV, which is also conserved in SARS-CoV-2, abolishes viral propagation in Vero-E6 cells[10]. In our assay, mutation of SARS-CoV-2 nsp12 R116 to a lysine or an alanine abolished any incorporation of radioactivity in nsp7 and nsp8 (Fig. 1c, Supplementary Fig. 1), further demonstrating the central role of this nsp12 RdRP protein in nucleotidyation. Promiscuous labeling of a non-relevant protein, bovine serum albumin (BSA), was not observed. Intermolecular labeling of nsp12 was also not observed. This suggested that nsp12-mediated nucleotidylation exhibits substrate specificity, although in vivo studies are needed to delineate the biological role of this nucleotidylation activity and to confirm these substrates.

### Identification of GMP-modified peptides using stable isotope labeling and LC–MS/MS.
We next utilized heavy isotope incorporation and the high mass accuracy of a tribrid Orbitrap based mass spectrometer to confirm that the transfer of radioactivity to the nsp proteins was a result of nucleotidylation. We performed nucleotidylation reactions in the absence of nucleotide, in the presence of natural GTP or, alternatively, GTP containing the heavy isotopes $^{15}$N or $^{13}$C. This allowed us to identify GMP-labeled peptide peaks using liquid chromatography (LC)–mass spectrometry (MS) alone, independent of MS/MS spectrum acquisition and matching. Unlabeled nsp peptides that were present in all sample injections ran at similar retention times (Fig. 2a, Supplementary Fig. 3). LC–MS peaks that corresponded to GMP-labeled peptides were identified by two criteria (Figs. 2b and 2c, Supplementary Table 1): (1) their absence in unlabeled sample injections, and (2) the presence of appropriately mass-shifted peptides at the same retention time in samples that were labeled with $^{15}$N- and $^{13}$C-GTP. Although a typical SARS-CoV-2 sample had ~13,000 different LC–MS peaks, only three peaks were found that met the above criteria of GMP-labeled peptides (Supplementary Fig. 4, Supplementary Data 1 and 2). Similarly, the sample set for EAV nsp proteins contained ~91,000 peaks, but only 50 peaks satisfied the specified criteria as GMP modified candidates (Supplementary Fig. 5, Supplementary Data 3).

For the next step of analysis, candidate peaks were subjected to MS/MS analysis in the tribrid mass spectrometer using higher energy collisional dissociation (HCD) fragmentation methodology as well as electron transfer/higher energy collision

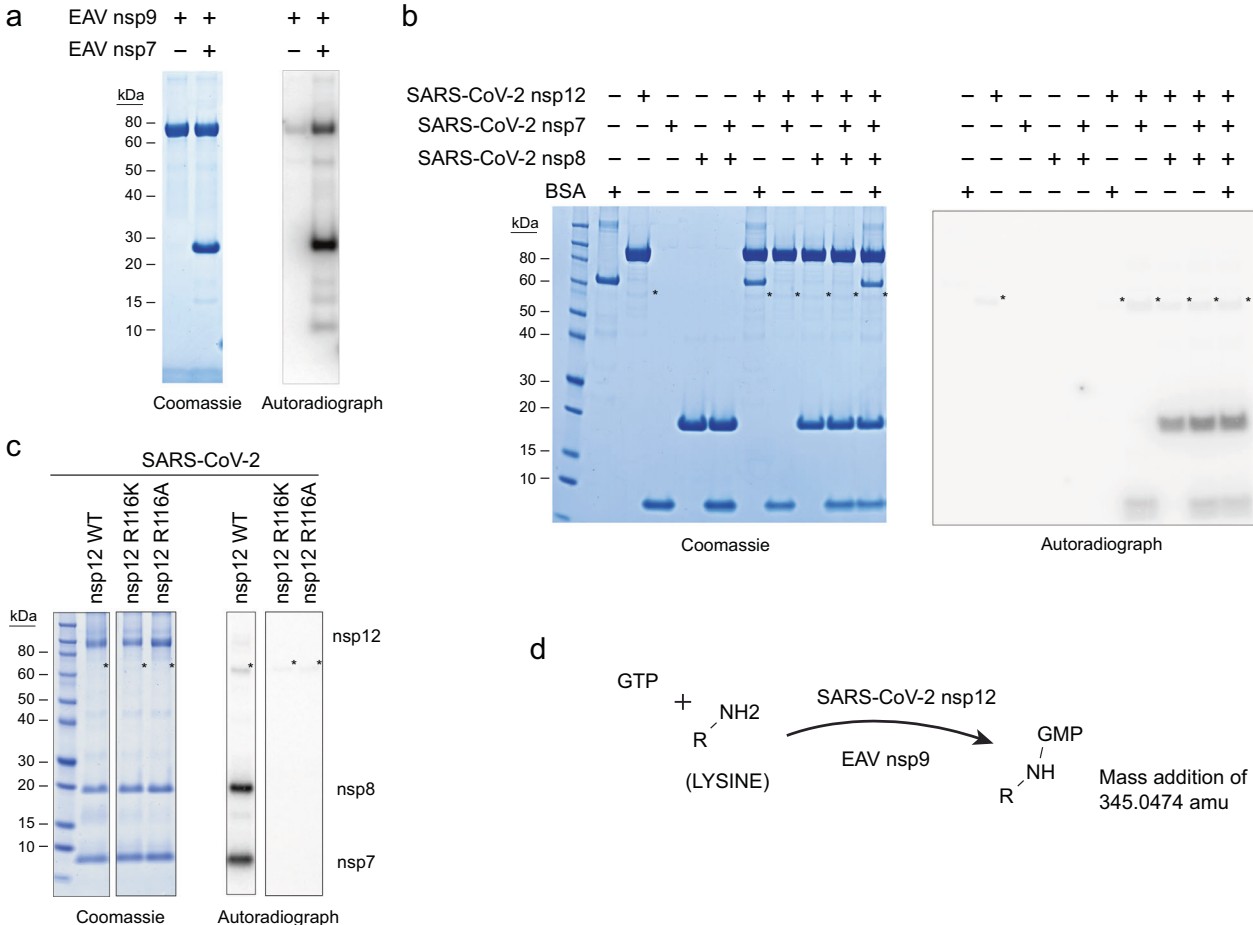

**Fig. 1 SARS-CoV-2 and EAV RdRP proteins are radiolabeled by incubation with α-$^{32}$ P-GTP. a** SDS-PAGE analysis of nucleotidylation reactions that contained EAV nsp9 with or without nsp7. **b** SDS-PAGE analysis of nucleotidylation reactions that contained SARS-CoV-2 nsp7, nsp8, and/or RdRP nsp12 combinations along with BSA as indicated. **c** SDS-PAGE analysis of nucleotidylation reactions that contained SARS-CoV-2 proteins nsp7 and nsp8 with nsp12 WT or indicated mutants. **d** Summary of general findings observed in this study. Coomassie stained gels and autoradiographs are shown to visualize total protein or proteins that were radiolabeled, respectively. Asterisks indicate the location of a faint band observed in both the Coomassie stained gel and autoradiograph. Detection of nucleotidylation of EAV and SARS-CoV-2 proteins by radiolabeling were observed in three separate experiments.

dissociation (EThcD). EThcD often preserves the attachment of labile peptide modifications that are difficult to preserve with HCD alone[20–22]. Table 1 summarizes all GMP-modified peptides identified by MS/MS using the Sequest HT database search algorithm (also see Supplementary Data 4)[23,24]. In SARS-CoV-2, we could only verify a single GMP-modified site by MS/MS, which was observed in multiple versions of the same peptide (Fig. 2d, Table 1, and Supplementary Table 2). For EAV, five total sites were identified in nsp proteins. Sites in EAV proteins included nucleotidylation of nsp7 lysine-143 (K143), which was observed in multiple peptides, and one site in nsp9 (Table 1, Fig. 3, and Supplementary Tables 3–5).

**MS/MS spectral characteristics of GMP-labeled peptides.** The most prominent feature in HCD spectra was the fragmentation of the GMP moiety itself that, in every instance, left a characteristic and dominant guanine ($C_5N_5OH_6$) tracer ion (Table 1, Fig. 3, Supplementary Tables 3–5 and Supplementary Fig. 6–10). The guanine ions were also observed in EThcD spectra, but at a low intensity. The mass of the guanine fragment changed appropriately depending on whether the peptide was labeled with GMP, $^{13}$C-GMP or $^{15}$N-GMP. The theoretical masses of guanine that is derived from each of these versions of GMP are 152.0572,

157.0740 or 157.0424, respectively. Peptide ions that do not contain the modification were prominent in these HCD spectra. For example, when the modification was located closer to the N-terminus, the y-ion series is easily identified, and the b-ion series is missing (Fig. 3a, b). Similarly, when the modification is closer to the C-terminus, the b-ion series is prominent, and the y-ion series is generally absent (Fig. 3c). The missing ion series was often observed with the loss of the GMP modification (Supplementary Fig. 6 and Supplementary Fig. 10), which makes its localization difficult if HCD is used alone. One main ion that has lost the GMP modification along with one or two $H_2O$ molecules often stands outs, for example, the 926.46 and 917.45 m/z ions in Figs. 3a and b, and the 971.96, 962.95 and 953.95 ions m/z in Fig. 3c.

Compared to HCD, EThcD spectra provided clean, higher-quality, and more interpretable data that had improved spectrum matching scores (Xcorrelation scores) and that contained both N-terminal and C-terminal fragment ions (c- ion series and z-, y-ion series, respectively) (Figs. 2c, 3, Supplementary Tables 3–5, and Supplementary Fig. 6–9). Preservation of the GMP attachment allowed better assignment of the modification site by the Sequest software but remained ambiguous in some cases. One example is the 1438.73035 ion in Fig. 2c. This m/z corresponds to the $z_{13}^+$ fragment that does not possess the GMP modification

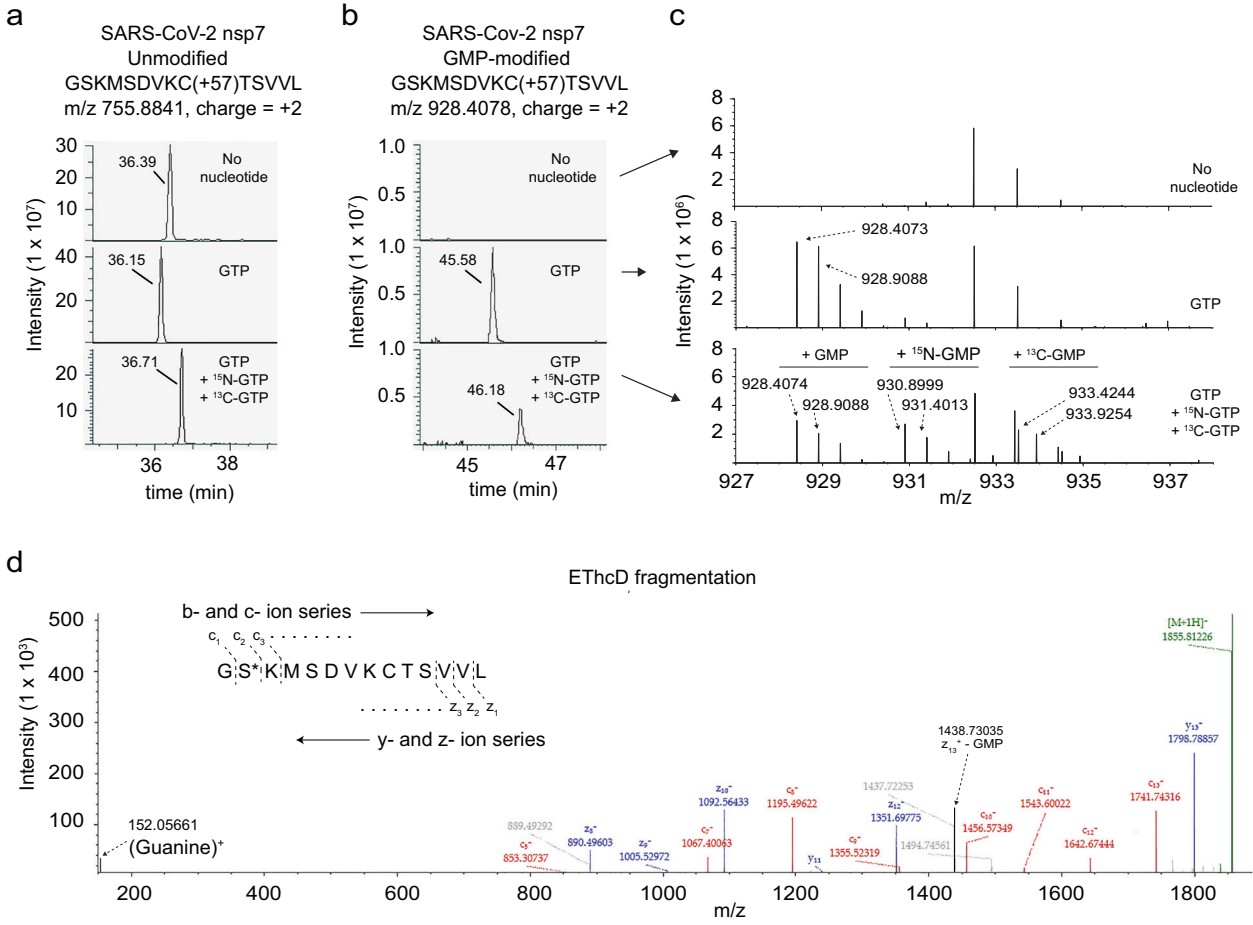

**Fig. 2 Detection of a GMP-modified SARS-CoV-2 peptide using LC–MS/MS. a** Extracted ion chromatograms at the indicated mass-to-charge ratio (m/z) (+/− 5 parts per million) show precursor peptide MS signals versus retention time for the unmodified SARS-CoV-2 nsp7 peptide 1–14 with a charge of +2. Signals are shown for samples that were incubated without GTP (no nucleotide), with GTP, or with a mixture of GTP, $^{15}$N-GTP and $^{13}$C-GTP. **b** MS signals at the m/z of 928.4078 for these same samples. This mass corresponds to the GMP-modified version of the same peptide with a charge state of +2. **c** The average mass spectrum across these peaks is shown from the m/z range of 927–938 in order to visualize the presence or absence of the isotopic mass envelops for the GMP-labeled peptide and the corresponding $^{15}$N- and $^{13}$C-labeled GMP peptides. For the unlabeled sample, average mass spectrum for retention times of ~45.68–45.78 is shown. **d** The top-scoring EThcD peptide spectrum match for same GMP-labeled peptide is shown with labeling of main fragment ions. Ion series fragmentation for this peptide is illustrated with the asterisk indicating the GMP-modification site. Red, blue and green coloring indicate c- fragment ions, z- and y- fragment ions, and precursor ions, respectively. The GMP-modified SARS-CoV-2 nsp7 peptide was observed in two separate experiments each with two technical replicates.

and would normally indicate the GMP modification is attached to the N-terminal glycine. However, a prominent $y_{13}^+$ ion at 1798.78857 that contains the GMP modification also was present, which indicated the modification was attached to serine-2. Attempts to make recombinant nsp7 without a glycine-1, which is the penultimate residue in the tobacco etch virus protease (TEV) cleavage site used to remove a N-terminal purification tag, abolished TEV cleavage and therefore failed. Top scoring GMP site assignments are listed in Table 1 (see also Supplementary Data 4).

**Mutational analysis of GMP-labeled protein sites.** Thus, to test whether the GMP modification assignments were correct, each predicted GMP-modified site was mutated to an alanine, one at a time. Peptides were then assessed by mass spectrometry for preservation of the GMP modification (Table 1, Supplementary Fig. 11) by looking for either MS/MS peptide spectrum matches, the predicted precursor mass in the MS scans, or diagnostic guanine 152.0572 ions that would be produced by HCD fragmentation of such GMP-containing precursor ions. Mutation of

EAV nsp7 K143, K156, K172 and nsp9 K380 eliminated the detection of the GMP modification on the corresponding peptide, thereby indicating the modification site prediction was correct. The unmodified version of each mutant peptide was readily detected. The peptide containing the EAV nsp7 mutation of threonine-3 to alanine (i.e., T3A mutation), however, was still modified at similar levels as wild type (WT), indicating that the transfer of the GMP moiety to the peptide was mediated via another residue. All EAV nsp mutants had similar levels of GMP radiolabeling, as expected, since nsp7 possessed multiple modification sites and other sites had yet to be identified (Supplementary Figs. 12–13 and Supplementary Data 3).

For SARS-CoV-2, mutation of nsp7 serine-2 to alanine (S2A) had minimal effect on either radiolabeling or the ability to observe the modified peptide by LC–MS/MS (Fig. 4, Table 1, and Supplementary Figs. 14–15). We also introduced a single K3A or double S2A-K3A mutation over concerns that a lysine-3 GMP-modification via a phosphormaide bond might be transferred to a more energetically favorable position on the neighboring residue during sample analysis[25–29] or that the initial assignment was incorrect. Mutation of these residues resulted in a reduction in

**Table 1 GMP-labeled peptides detected by LC–MS/MS.**

| Protein | Position[a] | Peptide sequence | Predicted modification site[b] | Modification site confirmed by neutralizing alanine mutation[c] | Top Xcorrelation score[d] | | HCD Guanine fragment ion[e] |
|---|---|---|---|---|---|---|---|
| | | | | | EThcD | HCD | |
| SARS-CoV-2 nsp7 | 1–14 | GSKMSDVKCTSVVL | S2 | No (S2), Yes (K3) | 3.87 | 1.63 | Yes |
| | 1–10 | GSKMSDVKCT | | | 3.29 | 2.01 | Yes |
| | 1–8 | GSKMSDVK | | | 2.62 | – | Yes |
| | 1–24 | GSKMSDVKCTSVVLLSVLQQLRVE | | | 6.10 | 4.59 | Yes |
| EAV nsp7 | 1–24 | SLTATLAALTDDDFQFLSDVLDCR | T3 | No | 3.25 | 3.18 | Yes |
| | 140–157 | GLPKGAQLEWDRHQEEKR | K4 | Yes | 7.83 | 3.12 | Yes |
| | 140–156 | GLPKGAQLEWDRHQEEK | | | 7.36 | 5.52 | Yes |
| | 140–151 | GLPKGAQLEWDR | | | 3.09 | 2.25 | Yes |
| | 144–157 | GAQLEWDRHQEEKR | K13 | Yes | 4.14 | – | Yes |
| | 157–173 | RNAGDDDFAVSNDYVKR | K16 | Yes | 5.73 | 3.65 | Yes |
| | 158–173 | NAGDDDFAVSNDYVKR | K15 | | 6.04 | 2.32 | Yes |
| EAV nsp9 | 369–381 | SNLQTATMATCKR | K12 | Yes | 2.90 | 2.49 | Yes |

"–" indicates that corresponding HCD spectra did not pass MS/MS search criteria.
[a]For SARS-CoV-2, positions are shown for the recombinant protein that contains an N-terminal glycine. The virally produced protein does not contain a glycine at position 1.
[b]Assigned GMP localization for the top-scoring spectrum match.
[c]For GSKMSDVKCTSVVL, mutation of lysine-3 to alanine, but not serine-2, led to substantial reduction of MS signal. Longer and shorter versions of peptide were not assessed. For assessment of K156 modification site, the corresponding chymotrypsin WT and mutant peptide were examined instead of the tryptic peptide that was originally observed (See Supplementary Fig. 11). Only peptide 140–151 was assessed for mutational analysis of the K143 modification site due to the presence of the second K156 site in the other peptides.
[d]Top score amongst GMP, [15]N-GMP and [13]C-GMP labeled peptides are shown.
[e]Indicates guanine fragment ion was the predominant ion in corresponding HCD spectra, even when they were not identified as peptide spectrum matches using the MS/MS search criteria.

the amount of nsp7 radiolabeling, which was confirmed using LC–MS/MS (Fig. 4). This data indicated that lysine-3 (lysine-2 in the natural SARS-CoV-2) is the preferred acceptor site for GMP. A small amount of residual GMP modification was still observed on the peptide even after mutation of lysine-3 (Supplementary Fig. 15). The lower MS signal made GMP localization impossible, but we hypothesize that GMP can be added to the N-terminus, albeit not full efficiency, based on the above evidence for amine-linked GMP attachment chemistry in the WT peptide.

**Protein nucleotidylation using UTP.** We next explored whether other nucleotides besides GTP could be used by SARS-CoV-2 or EAV NiRAN domains to mediate protein nucleotidylation. In a previous report, EAV was labeled more efficiently using uridine triphosphate (UTP), compared to GTP at a pH 8.5 with total nucleotide and protein concentration of 0.17 μM and 2.5 μM, respectively[10]. Using similar concentrations, we calculated that ~0.006–0.03% of SARS-CoV-2 or EAV proteins were labeled, which used approximately 0.06 to 0.3% of the nucleotide (Fig. 5a, Supplementary Fig. 16). However, protein labeling efficiency was approximately 1.0–4.8% at the higher, physiological nucleotide concentration of 200 μM[30]. The absolute efficiency of such in vitro biochemical reactions is affected by large number of factors (protein stability, lack of additional cofactors, buffer conditions, etc.)[31], making it difficult to draw conditions on how the efficiency observed in vitro directly relates to those in vivo. Surprisingly, we found α-[32]P-GTP labeled EAV nsp7 and nsp9 more efficiently than α-[32]P-UTP, both when the total nucleotide amounts were below or above the protein concentration (0.2 μM or 200 μM nucleotide versus 0.6–2.7 μM protein). A competition experiment with cold nucleotide (316 μM) to inhibit [32]P-GTP protein labeling confirmed that GTP outcompeted UTP, whereas adenosine triphosphate (ATP) and cytidine triphosphate (CTP) had little effect (Fig. 5b, c, Supplementary Fig. 17). In contrast, SARS-CoV-2 was preferentially labeled when using UTP, which was especially apparent at the higher, more physiological concentrations of nucleotide. Surprisingly, all four nucleotides had a significant inhibitory effect on [32]P-GMP covalent attachment with cold GTP being the most potent, followed by UTP, ATP and

CTP, respectively. It is unclear why UTP labeled SARS-CoV-2 proteins more effectively (Fig. 5a) but did not outcompete GTP for GMP protein labeling (Fig. 5b, c, Supplementary Fig. 17). A possible explanation is that resulting uridine monophosphate (UMP) covalent attachment is more stable. LC–MS/MS revealed that UMP was added to the same SARS-CoV-2 nsp7 peptide, which therefore appeared as a mass addition of 306.0253 atomic mass units (amu) (Fig. 5d, e, Supplementary Table 6). Consistent with the observed increase in UTP labeling efficiency, the UMPylated-peptide signal was much stronger in these samples, compared to GMP labeled samples. Although HCD fragmentation was performed, a striking diagnostic ion for UMP modification on this peptide could not be discerned. EThcD fragmentation yielded MS/MS spectra that had higher peptide match scores by the Sequest search software compared to HCD. Localization analysis indicated that UMP was also added to the serine-2 or lysine-3, potentially being the first time UMPylation has been observed on these sidechains. These results indicated that UTP could also be used as a nucleotidylation substrate by SARS-CoV-2 nsp12.

## Discussion

Until now, successful MS identification of a GMP-peptide adduct has not been demonstrated. The results reported here allow a more systematic and high-throughput method to find such sites. We show that EThcD fragmentation is favorable over HCD fragmentation because both sets of the N- and C-terminal ion series are generated with the modification intact. Much of the energy provided in HCD fragmentation causes destruction of the GMP molecule, similar to what occurs with glycan post-translational modifications in the collisional-induced dissociation (CID) fragmentation technique that is comparable to HCD[32,33]. However, HCD can be a powerful diagnostic tool used to locate peptides that are GMP-modified because, without exception, the dominant MS/MS ion is always a guanine at a m/z of 152.0572. Adenosine monophosphate (AMP)-peptide adducts have been examined by MS/MS using CID[34]. These spectra also featured destruction and loss of the nucleotide modification that could make peptide identification difficult. However, fragmentation of

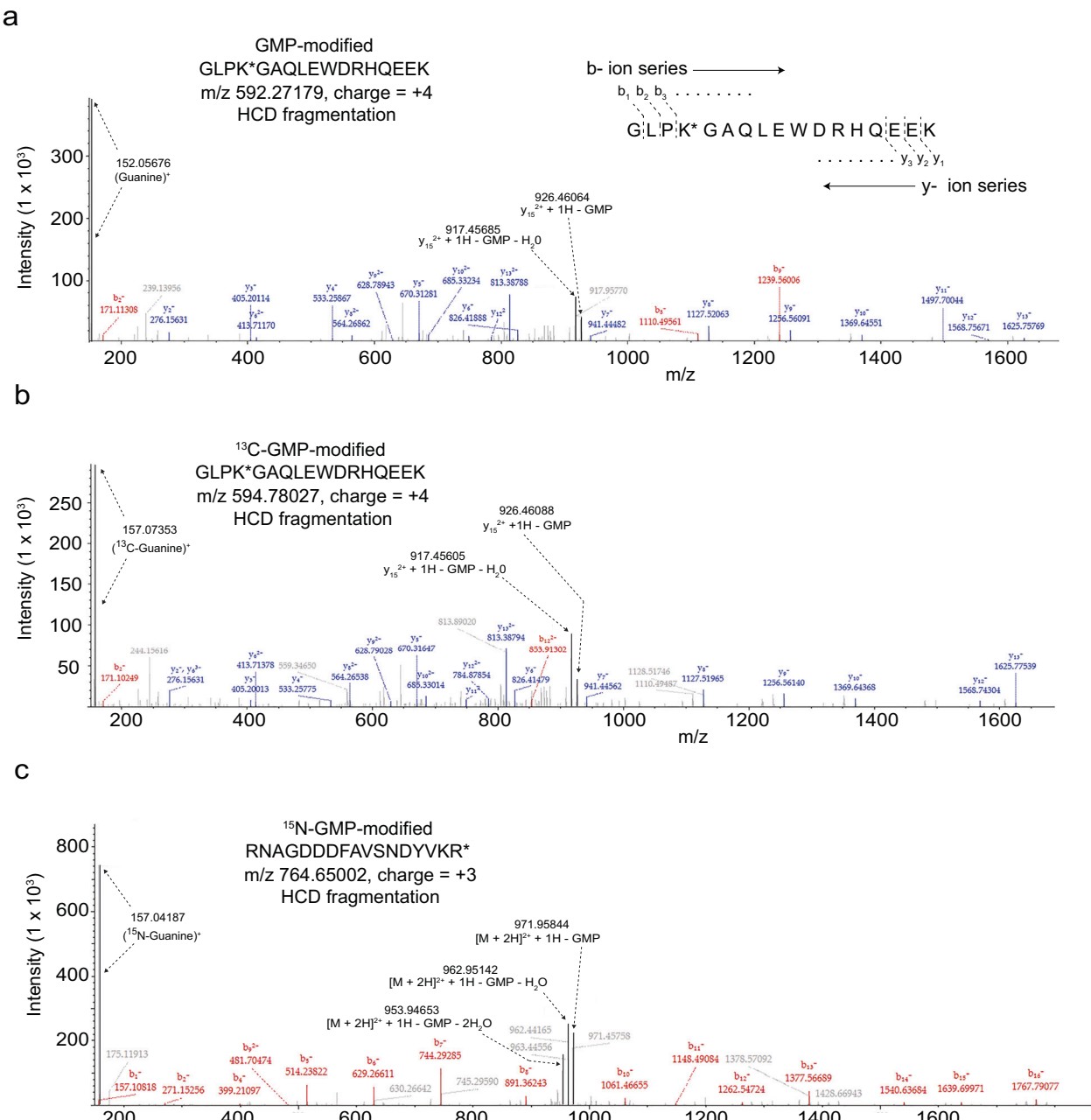

**Fig. 3 MS/MS spectra of GMP, $^{15}$N-GMP and $^{13}$C-GMP modified peptides fragmented with HCD. a** Top-scoring HCD peptide spectrum match for an EAV nsp7 peptide that is labeled close to the N-terminus with GMP. Ion series fragmentation for this peptide is illustrated. **b** Top-scoring HCD peptide spectrum match from an EAV nsp7 peptide that is labeled close to the N-terminus with $^{13}$C-GMP. **c** Top-scoring HCD peptide spectrum match from an EAV nsp7 peptide that is labeled close to the C-terminus with $^{15}$N-GMP. Note that Sequest HT assigned the $^{15}$N-GMP modification to R17, instead of the correct site at K16. Spectra are additionally labeled with guanine fragment ions and dominant peptide fragments that have lost the GMP modified plus one or two $H_2O$ molecules. Asterisks indicate the assigned GMP-modification site. Red and blue coloring indicate b- fragment ions and y- fragment ions. The GMP-modified EAV peptides were observed at least two separate experiments.

the AMP moiety left diagnostic ions at 136.1 and 250.1 that could be used to locate modified peptides in complex mixtures.

Even with EThcD fragmentation, the modification site assignment was not always correct. For SARS-CoV-2 nsp7, mutagenesis was required to identify lysine-3 as the critical residue responsible for the GMP attachment. Many factors could be responsible for this including minor loss of the GMP modification during fragmentation apparent by residual guanine ion fragments, the potential for phosphorous-linked modifications to be shuffled in the fragmentation process[25–29], the possibility of

more than one peptide isoform existing, or insufficient peptide quantities. This same possibility may exist for the EAV N-terminal peptide that appeared to be modified at threonine-3, whose mutation to alanine indicated it was not essential for the initial transfer of GMP to the peptide. How the GMP is transferred to this peptide is unclear, but our data suggests formation of a phosphoramidate via an amine is necessary, suggesting attachment could occur via the N-terminus. It should be noted that non-canonical phosphorylation on basic amino acids, which uses identical phosphoramidate chemistry (i.e., the P–N bond) for

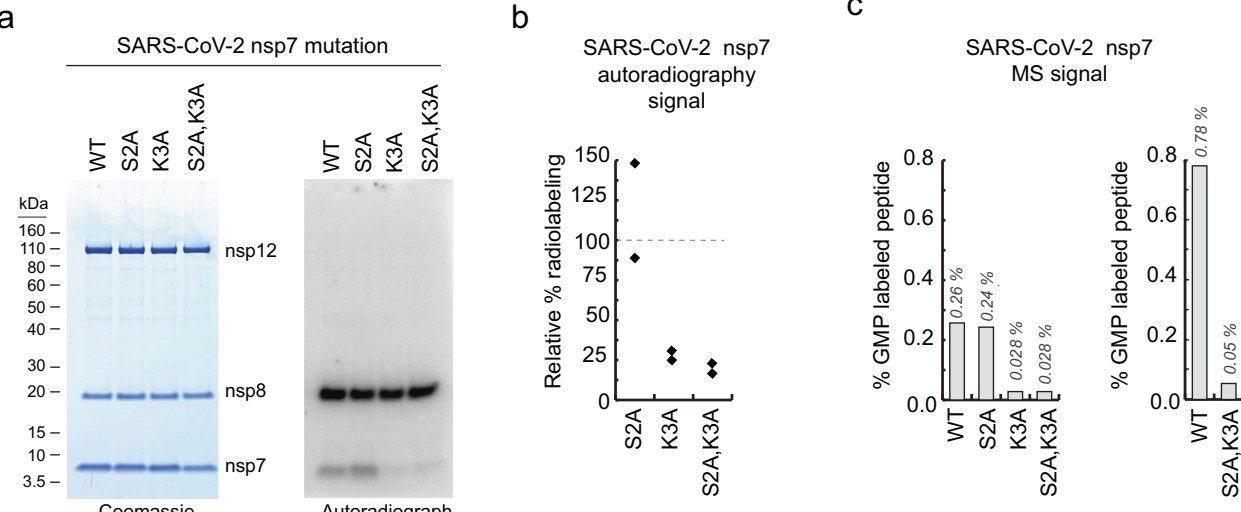

**Fig. 4 GMP labeling of SARS-CoV-2 nsp7 mutants. a** Radiolabeling of nsp7 was assessed after mutating indicated residues to alanine. A Coomassie-stained SDS-PAGE gel and an autoradiograph are shown. **b** Quantitation of mutant radiolabeling relative to WT in two separate experiments normalized to nsp8 signal. Individual datapoints (black diamonds) are shown (n = 2). Data are scaled to WT signal that was set to 100 percent, as indicated by the dotted line. **c** LC–MS precursor ion quantification of percent (%) GMP modification of the SARS-CoV-2 peptide 1–14 upon introduction of indicated mutations in comparison to total peptide amount (unmodified and modified). Each bar represents a single data point. Two experiments were performed for WT and for the S2A, K3A mutation. A single experiment was run for the nsp7 S2A and nsp7 K3A mutant proteins.

the GMP linkage, is beginning to emerge as an important component of signaling systems in nature[21,35]. These phosphoramidate bonds are more labile under the conditions commonly used in analytical instrumentation such as high-pressure liquid chromatography and mass spectrometry and are therefore understudied. This highlights the need for more comprehensive studies that examine the impact of additional factors on their in vivo and in vitro lability, such as the specific local chemical environment provided by unique amino acid sequences on either side of the modified residue.

Multiple studies have now indicated that the NiRAN domain binds nucleotide and is responsible for nucleotidylation activity[10,19,36]. In SARS-CoV-2, the NiRAN domain (residues 1–250) is distant from the RNA polymerase active site (residues 400–932) and also the binding sites for nsp7 and nsp8[8,9]. However, in an aqueous solution, protein complexes are in constant dynamic equilibrium with their monomeric states, and the ordered steps of the multimeric RdRP assembly and rearrangement throughout the viral life cycle is unknown. Thus, in this study, nsp12 likely modified nsp7 and nsp8 in an unbound state, although it cannot be ruled out that nucleotidylation observed occurred in trans between different nsp12/nsp8/nsp7complexes[8,9,11,19,36,37]. SARS-CoV-2 nsp7 is not found outside of the Coronaviridae family (Supplementary methods, Supplementary Fig. 18)[38]. Within the Coronavirinae subfamily, lysine-2 is well-conserved except in common moorhen coronavirus HKU21 and Bat coronavirus HKU9-10-2, where an arginine or asparagine are found, respectively. In nsp7, 23 of 83 residues show conservation, which indicates lysine-2 has structural or functional importance. The SARS-CoV-2 RdRP structure shows nsp7 lysine-2 is located at in interface between nsp8 and nsp12, thus could possibly modulate their association.

For EAV, cofactor binding may activate nucleotidylation activity since the presence of nsp7 increased GMP radiolabeling substantially (Fig. 1a). The EAV nsp9 nucleotidylation site at K380 is highly conserved across nidoviruses (Supplementary methods, Supplementary Fig. 19). The homologous residue in the SARS-CoV-2 RNA polymerase nsp12 is K545, which is located in the NTP binding site for the RNA polymerase domain that contacts

and stabilizes the bound nucleotide for incorporation into RNA (Supplementary Fig. 20)[8,31,36]. Thus, it is possible nucleotidylation of this residue could alter RNA synthesis efficiency.

EAV nsp7 is only found in Arteriviridae family. The confirmed nucleotidylation sites in EAV nsp7 (K143, K156 and K172) are in located the nsp7β portion of the protein (residues 124–225) that is more poorly conserved in Arteriviridae family (Supplementary methods, Supplementary Fig. 21) and may be separated from the nsp7α portion (residues 1–123) in vivo by peptidase cleavage[17,39,40]. While K143 is not found outside of EAV, both K156 and K172 are also found in porcine reproductive and respiratory syndrome virus (Supplementary Fig. 21). The lack of any structure for nsp7β makes it difficult to speculate how modification of these residues may affect function.

Nucleotidyl transferase activity plays a role in diverse biological processes in bacteria, viruses and eukaryotes. The addition of AMP (AMPylation or adenylation) was first described as a mechanism to regulate E. coli glutamine synthetase activity[41]. It is now recognized, along with UMPylation, as a post-translation modification that impacts bacterial DNA replication[42,43], ER and oxidative stress responses[44–46], bacterial pathogenesis[47,48], and viral RNA synthesis[49] where AMP nucleotides are stably attached to serine, threonine or tyrosine resides[50,51].

In SARS-CoV-2, a recent report demonstrated nsp12-mediated GMP transfer to an RNA substrate in the presence of the RNA triphosphatase nsp13, i.e., the first successful RNA capping (guanylate transferase) reaction observed for nidoviruses[11]. Introduction of a nsp12 K73A mutation eliminated this activity, indicating that the NiRAN domain is responsible. Presumably, nsp13 removed a phosphate group from the 5′ RNA position, which thereby allowed the NiRAN domain of nsp12 to transfer the GMP moiety to the 5′ diphosphate adenosine. Protein nucleotidylation was not observed. However, acidic conditions were used throughout the mass spectrometry sample prep, which may have led to destruction of the potentially labile modification (Supplementary Fig. 2).

Overall, it is not yet clear how this in vitro assay relates to the physiological mechanism in infected host cells. For instance, how does an RNA molecule being synthesized in the distant part of the

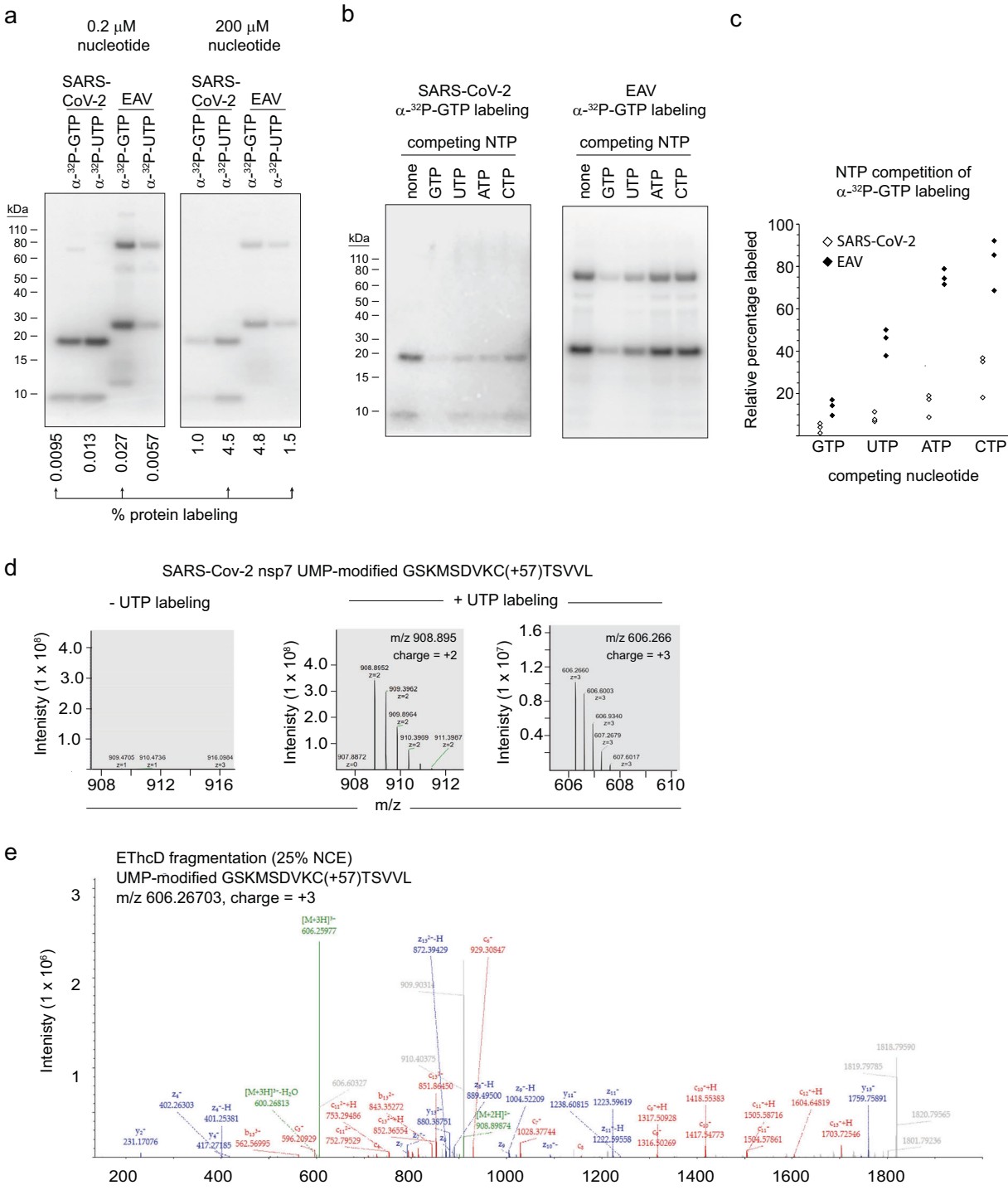

**Fig. 5 SARS-CoV-2 and EAV radiolabeling with UMP. a** Radiolabeling of SARS-CoV-2 (nsp7 and nsp8) or EAV (nsp7 and nsp9) proteins using either α-³²P-GTP or α-³²P-UTP at a total nucleotide concentration of ~0.2 μM or 200 μM. The total percentage of nsp proteins labeled in each experiment is shown below autoradiographs, given 0.6–2.7 μM of each labeled protein per reaction. Identical exposures are shown for left and right panels. Results represent a single experiment. **b** Competition of α-³²P-GTP labeling (at 10 μM) with 316 μM of the indicated cold nucleotide. Autoradiographs are shown of an SDS-PAGE protein analysis. **c** The percentage ³²P-GMP labeling remaining after direct competition with 316 μM GTP, UTP, ATP or CTP. Three separate experiments (*n* = 3) were performed for both SARS-CoV-2 (white diamonds) and EAV (black diamonds), and individual data points are shown. **d** LC–MS spectra at 39.7 min for SARS-CoV-2 samples that were labeled with UTP or not. Spectra show peaks at m/z of ~908.897 and 606.267, which correspond the SARS-CoV-2 nsp7 1–14 peptide (GSKMSDVKCTSVVL) in charge states of +2 and +3, respectively, that has been modified with UMP (306.0253 amu mass addition). **e** Top peptide spectrum match for UMP-labeled GSKMSDVKCTSVVL peptide that was derived from chymotrypsin-digested SARS-CoV-2 nsp7 protein. EThcD fragmentation of the 606.266 peptide ion was used with 25% normalized supplemental collisional energy (NCE). Detection of the UMP-modified peptide was observed in a single experiment with two technical replicates.

nsp12 molecule reach the NiRAN active site? Or is GMP transfer regulated differently under the limited concentrations of substate present during in vivo conditions? Traditionally, guanyl transferases involved in RNA capping form a GMPylated protein intermediate before movement of the nucleotide to the RNA[12–14]. In other RNA viruses, a uridylylated protein, known as VPg, is used to initiate RNA synthesis[52–57]. SARS-CoV-2 nsp7 and nsp8 are positioned to interact with the RNA 5′ terminus when bound to nsp12[36,37] and are critical for robust RNA replication[15,58]. Thus, although we do not have evidence that nsp12 mediated nucleotidylation of nsp7 or nsp8 plays a role in this process, the precise in vivo mechanism for this novel viral RNA capping system remains to be demonstrated. The methodology presented here is expected to be important in elucidating the exact biochemical steps taken that lead to RNA capping in SARS-CoV-2 and other organisms that use such GMP-protein capping intermediates. Alternatively, this study may reveal a new protein modification that requires further examination in vivo to elucidate its biological impact.

## Methods

**Expression and purification of the EAV and SARS-CoV-2 proteins**. DNA for SARS-CoV-2 nsp12 encompassing the a.a. 1–932 and EAV nsp9 1–693 was synthesized with codon optimization (Genscript) and cloned into pFastBac with an N-terminal MG addition and C-terminal TEV protease site and two Strep tags. Expression was performed by transducing recombinant baculoviruses into Sf21 insect cells (Expression Systems). Cells were harvested by centrifugation and resuspended in 25 mM HEPES pH 7.4, 300 mM sodium chloride, 1 mM magnesium chloride, and 2 mM dithiothreitol (DTT). The resuspended cells were then lysed using a microfluidizer, clarified by centrifugation at $25,000 \times g$ for 30 min and filtered using a 0.45 μm filter. RdRP proteins were purified using Streptactin Agarose (IBA) eluting with 2 mM desthiobiotin. Eluted protein was further purified by size exclusion chromatography using a Superdex200 column (GE Life Sciences) in 25 mM HEPES pH 7.4, 300 mM NaCl, 0.1 mM magnesium chloride, and 2 mM tris (2-carboxyethyl)phosphine. Full-length, codon-optimized nsp7 and nsp8 genes were cloned into pET46 for expression in E. coli. The N-terminal tags for SARS-CoV-2 nsp7 and nsp8 are MAHHHHHHVDDDDKMENLYFQG and for EAV nsp7 is MAHHHHHHVGTENLYFQ. The TEV protease cleavage sites (ENLYFQ|G) were positioned to leave N-terminal glycines on SARS-CoV-2 nsp7 and nsp8 and no additional amino acids on EAV nsp7. Proteins were expressed in Rosetta2 pLysS E. coli (Novagen). Bacterial cultures were grown to an OD600 of 0.8 at 37 °C, and then the expression was induced with a final concentration of 0.5 mM of isopropylβ-D-1-thiogalactopyranoside and the growth temperature was reduced to 16 °C for 14–16 h. Cells were harvested by centrifugation and were resuspended in 10 mM HEPES pH 7.4, 300 mM sodium chloride, 30 mM imidazole, and 2 mM dithiothreitol. Resuspended cells were lysed using a microfluidizer. Lysates were cleared by centrifugation at $25,000 \times g$ for 30 min and then filtered using a 0.45 μm filter. Protein was purified using Ni-NTA agarose (Qiagen) eluting with 300 mM imidazole. Eluted proteins were digested with 1% (w/w) TEV protease. TEV protease-digested proteins were passed over Ni-NTA agarose to remove uncleaved proteins and then further purified by size exclusion chromatography using a Superdex200 column (GE Life Sciences) in 25 mM HEPES pH 7.4, 300 mM sodium chloride, 0.1 mM magnesium chloride, and 2 mM dithiothreitol.

**Nucleotidylation assay**. Nucleotidylation assays were performed under native conditions similarly to those previously described[10]. The EAV nsp9 (0.7 μM) or SARS-CoV-2 nsp12 (0.6 μM) RdRP proteins were incubated with a 2–4 molar excess of EAV nsp7 (1.4 μM) or SARS-CoV-2 nsp7 (2.7 μM) and nsp8 (1.4 μM) proteins, respectively, in following buffer: 50 mM Tris pH 8.5, 6 mM MnCl₂, 1 mM DTT, 25 mM NaCl and 2% glycerol. Stock protein concentrations were determined by Bradford assays (Biorad). Labeling reactions were initiated by the addition of GTP and were incubated for 30 min at 30 °C. Radioactivity assays employed 0.5–5 μCi α-³²P-GTP or α-³²P-UTP (Perkin Elmer) with total nucleotide concentration of 0.17 μM except where indicated otherwise, whereas mass spectrometry assays used 0.2–1 mM GTP, ¹⁵N-GTP (Sigma), ¹³C-GTP (Sigma) or UTP. For competition experiments,10 μM α-³²P-GTP was mixed with 316 μM of cold ATP, GTP, UTP, CTP (Fig. 5b and c) (ThermoFisher) or the concentrations indicated (Supplementary Fig. 17).

**SDS-PAGE and autoradiography**. Samples were mixed with 4x LDS loading dye (Thermo) containing 200 mM DTT and analyzed on Bolt Bis-Tris Plus gels (Thermo) according to manufacturer's instructions. Gels were either stained with GelCode™ Blue Protein Stain (Fisher Scientific) or prepared for autoradiography by incubation in a 10% polyethyleneglycol 8000 (Sigma) solution for 30 min and gel drying at 70 °C for 45 min using a Hoefer gel drier, Savant condenser unit and

an TRIVAC pump (Leybolt). Gels were exposed to phosphor screens for 2 hour to 7 days and developed with a Molecular Dynamics Storm 850 imager. Gel images were examined using ImageJ opensource software. For quantitation, a standard curve was created by dotting known amounts of radioactivity ($5 \times 10^{-1}$ to $5 \times 10^{-5}$ μCi) on nitrocellulose and subjecting it phosphorimaging. Gel band and standard signal volumes were determined, the background was subtracted, and relative or absolute quantities were determined in Microsoft Excel after linear curve fits of a standard volumes (log (volume) vs log (μCi radioactivity)) that yielded an $r^2 > 0.99$.

**LC–MS/MS**. After nucleotidylation reactions were completed, they were brought to 6.7 M Urea in 50 mM ammonium bicarbonate, 5 mM DTT and incubated at 42 °C for 15 min. Cysteines were alkylated by the addition of 15 mM iodoacetamide for 30 min at room temperature. Following the neutralization of unreacted iodoacetamide with 15 mM DTT in order to prevent overalkylation[59], nsp proteins were diluted to a final concentration of 1 M urea and digested overnight with either Trypsin/LysC mix, Chymotrypsin, or GluC (Promega) according to manufacturer's instructions. The resulting peptides were desalted using OMIX C18 pipette tips (Agilent Technologies) or 1 mL C18 Sep Pak cartridges (Waters) in 10 mM ammonium formate at pH 7.0, eluted in 75% acetonitrile, and dried to completion with a vacuum centrifuge.

For LC–MS/MS analysis, each digested sample was suspended in 0.1% formic acid or preferable 10 mM ammonium formate (pH 7.0) and maintained at 7 °C until analysis within 12 hours. Samples were loaded onto Thermo Scientific Easyspray C4 or C18 nanocolumn and eluted with a gradient to 80% acetonitrile / 0.1% formic acid at 300 nL/min at room temperature using an Ultimate 3000 series liquid chromatography system. Eluted peptides were ionized in-line with a Thermo Scientific Easy Spray source for direct analysis with a Thermo Scientific Fusion Lumos Orbitrap mass spectrometer and subjected to a targeted or data-dependent MS/MS acquisition scheme that collected both HCD and EThcD spectra in the high resolution orbitrap.

LC–MS peaks were defined by analyzing the data with the feature mapping and precursor quantification nodes in Proteome Discoverer 2.4 that determined the retention time, charge state, and abundance of every peak in each searched file. The data were filtered and exported to Microsoft Excel to search for peaks that (1) were not present in unlabeled samples and (2) that possessed two peaks with the same charge state at a similar retention time (+/−0.5 min) and were mass-shifted by 4.9852 or 10.0336 amu [MH + ion masses] for ¹⁵N- or ¹³C-GMP labeling, respectively, within a +/− 1.5 parts per million error limit (Supplementary Fig. 4–5). Supplementary Data 1 Supplementary Data 2, and Supplementary Data 3 are Microsoft Excel worksheets used to perform these searches once the data was exported.

To identify peptide spectrum matches, data were searched with the Sequest HT algorithm through Proteome Discoverer 2.4 and matched with a mass error tolerance of 0.04 Da for MS/MS fragment ions[24]. GMP or UMP adducts were assigned by allowing a dynamic modification for the mass addition of 345.0474 (and 350.0326 or 355.0810 for ¹⁵N and ¹³C labeled GMP) or 306.0253 amu, respectively, to H, S, T, Y, K or R, based on known phosphodiester or phosphoramide attachment chemistry[12,34,60], although all possible sites were considered in initial, preliminary searches. For the SARS-CoV-2 GMP localization on nsp7, initial searches assigned the modification site to the N-terminal glycine at peptide position 1, but the modification site was re-assigned manually to ser-2 (ser-1 in natural nsp7) because of the presence of the y₁₃₊ ion (ser-2) at an m/z of 1798.78857 that contained the GMP moiety. For each unique peptide m/z, the top-scoring peptide spectrum matches for both EThcD and HCD are presented in Supplementary Data 4 along with GMP-attachment site assignments and additional information. The top-scoring, overall match for each peptide is presented in Table 1 along with the corresponding attachment site assignment. All matches had a precursor mass error of +/− 1.5 parts per million or less and were validated with a target-decoy or Percolator search using a cut-off false discover rate value of 1%. Only peptides with a top spectrum match Xcorrelation score of 2.5 or greater were reported. Extracted ion chromatograms and spectra were generated using Freestyle software (Thermo Scientific). Note that for EAV reactions, not all GMP-modified candidate peaks were examined in MS2, because all EAV data was acquired prior to data analysis and MS2 was performed in a data-dependent acquisition scheme. Though a single site was found in SARS-CoV-2, additional phosphoramide linked GMP sites likely exist that may not have been observed because of the lability of the GMP attachment in the acidic conditions used for LC–MS/MS (Supplementary Fig. 2). Percent GMP modification was calculated using LC–MS areas obtained with Freestyle software (Thermo) (GMP-modified peptide area divided by the total sum of the GMP-modified and unmodified peptide area). Additional LC–MS methods describing the analysis of mutant proteins are described in Supplementary methods.

**Reporting Summary**. Further information on research design is available in the Nature Research Reporting Summary linked to this article.

## Data availability.

The raw mass spectrometry datasets and proteome discoverer results that were generated and analyzed in the current study are available through the MassIVE repository under the identifier MSV000085857. All other data are available within the paper,

the Supplementary Information file, the Supplementary Data 1–4 files, or can be provided by the corresponding author upon reasonable request.

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

## Acknowledgements

Funding to support this work was provided to R.N.K. from NIH/NIAID (Grant No. AI123498) and M.R.S. from NSF (Grant No. MCB-1713899).

## Author contributions

B.J.C. designed and performed the experiments and wrote the manuscript. A.S.L. expressed, purified, and provided the proteins. R.N.K. purified and provided the proteins, conceived the project, helped to design experiments, and contributed to the writing of the manuscript. M.R.S. conceived the project, helped to design experiments, and wrote the manuscript.

## Competing interests

The authors declare no competing interests.
