## [Peer Review File · Communications Chemistry]

Reviewers' comments:

Reviewer #1 (Remarks to the Author):

The manuscript by Conti et al. reports arterivirus (EAV) nsp9 and coronavirus (SARS-CoV-2) nsp12-mediated protein guanylation. While autoguanylation activity has previously been shown for a recombinant EAV nsp9 (NiRAN-RdRp) (Lehmann et al., 2015), guanylation of other viral proteins by nidovirus NiRAN domains has not been demonstrated previously. Using [α - 32 P]-GTP, the authors now show that, in the presence of EAV nsp9, the EAV nsp7 protein is radiolabeled. Similarly, SARS-CoV-2 nsp12 (but not the K73A NiRAN active-site mutant) is shown to transfer GMP to two recombinant viral proteins (nsp7 and nsp8). Both EAV nsp9 and SARS-CoV-2 nsp12 failed to GMPylate bovine serum albumin which was used as a specificity control in the nucleotidylation assay. To identify GMP-protein adducts produced in these reactions, the authors used mass spectrometric technologies (along with GTP-containing N-15 and C-13 heavy isotopes). LC-MS peaks of candidate GMP-labeled peptides could be identified and were subsequently subjected to MS/MS analysis by HCD and EThcD. In total, 5 GMP-modified sites were identified for EAV (4 of which were assigned to EAV nsp7 and 1 to nsp9). One GMP adduct (with nsp7 Ser-1 being modified) was identified for SARS-CoV-2. No GMP adducts were identified for SARS-CoV-2 nsp8, although there was convincing evidence for GMPylation of this protein in the nucleotidylation assay using radiolabeled [α - 32 P]-GTP. Overall, the manuscript is well written and the data provide preliminary evidence that nidovirus NiRAN domains use the (previously reported) NiRAN(-RdRp)-associated nucleotidyl transferase activity to modify other viral proteins. Although the study provides convincing evidence for arterivirus nsp9- and coronavirus nsp12-mediated GMPylation of other proteins, this reviewer feels that the biological relevance of the *in vitro* observations obtained for recombinant proteins remains unclear. Given that the "substrate" proteins used (EAV nsp7 and SARS-CoV-2 nsp7 and 8) are known to interact tightly with the cognate NiRAN-RdRp proteins, additional data should be provided to (i) corroborate the specific modification of specific (conserved or non-conserved?) residues in these proteins and (ii) measure the transfer efficiency of the reaction. As a minimum, mutant forms of (EAV and SARS-CoV-2) nsp7 and SARS-CoV nsp12 with substitutions of residues that are suggested to be GMPylated should be included in the study. Also, given some remaining doubts (page 6, 102f.) regarding GMPylation of the extra Gly-1 versus the natural Ser (Ser-2 in the recombinant protein) it would be desirable to produce and analyze an nsp7 protein with an authentic N-terminal Ser-1, along with mutant proteins in which the Ser proposed to carry an O-linked (rather than N-linked) GMP is replaced with Ala. Also, the conservation in other arteri- and coronaviruses of residues identified to be GMPylated should be discussed.

Other points:

1. Please provide information on the protein concentrations used in the nucleotidylation assay (page 18).
2. Given that Lehmann et al. reported that EAV nsp9 autonucleotidylation is more efficient with UTP compared to GTP, I suggest that experiments using UTP be performed.
3. Page 4, 63f. The paper by Chen et al (Cell 2020, 182(6):1560-1573.e13. doi: 10.1016/j.cell.2020.07.033. Epub 2020 Jul 28) should be cited in the context of the presumed nucleotidyl transferase active site.

Reviewer #2 (Remarks to the Author):

The authors described the detection and identification of GMP-modified amino acids in nidovirus RdRP-associated proteins using heavy isotope-assisted MS and MS/MS peptide sequencing. They were able to identify several *in-vitro* nucleotidylation sites by incubation of high dosage of nucleotides. Although they failed to identify *in-vivo* sites which were detected by radio-isotope, the mass mapping method they described is logical and comprehensive. Their results are useful for

detecting GMP-peptide linkages which are still lacking in the current methods. Although this work is interesting, some concerns regarding experimental procedures are in below.

1. The authors have used nucleotides which concentration is higher than that for radio-labeling by near 3 orders of magnitude for in-vitro nucleotidylation reaction. It is not clear why nucleotidylation reactions were still performed under denatured conditions, 6.7 M Urea in 50 mM ammonium bicarbonate and 5mM DTT. This made the whole reaction condition greatly deviate from the cellular condition. I suggest to incubate high dosages under native conditions.
2. It is not clear why 15 mM DTT was used to neutralize unreacted iodoacetamide. Most protocols use DTT to break (reduce) disulfide linkages followed by alkylation rather than quenching extra iodoacetamide as they used. Explanations of their rationales are required.

Major changes to COMMSCHEM-20-0326-T by Conti *et al.*:

- 1) Predicted GMP modification sites have been mutated to alanine. We also introduced the K3A and S2AK3A mutation into SARS-CoV-2 to test lys-3 attachment over concern that the GMP modification site could shuffle during analysis as discussed in the paper. We then asked if the corresponding peptides were still modified by GMP.

We were able to validate the following GMP attachment sites by the disappearance of the peptide modification upon introduction of the mutation: EAV nsp7 K143, K156 and K172, EAV nsp9 K380.

For the SARS-CoV-2 nsp7 peptide, the experimental data indicated the lys-3 was critical in mediating GMP attachment because the amount of modified peptide is reduced 80-90% when the corresponding alanine mutation is introduced. We found that EAV nsp7 T3 was not essential for GMP modification of the N-terminal peptide.

Our language throughout the manuscript has been adjusted accordingly. See lines 8-10, lines 124 – 146, lines 186 - 196, Table 1 and the additional Figure 4 for these changes. Supplementary Figures 10 – 12 have also been added to support these findings.

- 2) We examined labeling of the EAV and SARS-CoV-2 proteins with UTP as well as GTP. Lehmann *et al.* found that UTP could also label EAV nsp9 (Nucleic Acids Res. 2015 Sep 30;43(17):8416-34). We found similar, but not identical results. In our hands, UTP labeled SARS-CoV-2 nsp7 and nsp8 more effectively, but GTP labeled the EAV nsp7 and nsp9 proteins better than UTP. We then performed a competition experiment with non-radioactive nucleotides (ATP, UTP, GTP and CTP) to examine which outcompeted radioactive GTP labeling most effectively. Consistent with the Lehmann *et al.* paper, ATP and CTP did not compete for EAV protein GTP labeling. In contrast, all nucleotides competed for SARS-CoV-2 protein labeling. Finally, we continued further and demonstrated the same peptide in SARS-CoV-2 was modified by UTP labeling, which resulted in attachment of a UMP molecule as verified by LC-MS/MS.

See lines 147 - 171 and the additional Figure 5 for these changes. Supplementary Figure 13 has also been added to support these results.

- 3) We looked at EAV and SARS-CoV-2 protein radiolabeling at more physiological GTP and UTP conditions (200 μ M), in addition to those conditions originally used by Lehmann *et al.* (\sim 0.2 μ M). The observations remained generally the same. We also provide as estimate of protein labeling efficiency.

See lines 147 – 152 and Figure 5a for this new data.

- 4) We added further discussion on the potential physiological significance of our findings. Namely, this enzymatic activity is believed to be a critical step in RNA capping as recently reported by Yan *et al.* (Cell. 2020 Nov 14;S0092-8674(20)31533-6. doi: 10.1016/j.cell.2020.11.016), though the detailed mechanism is unclear. Because protein nucleotidylation is traditionally an essential step in this enzymatic activity (reviewed in Nucleic Acids Res. 2016 Sep 19;44(16):7511-26. doi: 10.1093/nar/gkw551), which was not shown by Yan *et al.*, we believe our study will have great value in solving the exact mechanism of SARS-CoV-2 RNA capping in the future. We now discuss this possibility in the revised manuscript. In addition, discussion about the conservation of the GMP-modified residues has been added to the paper.

See lines 44 – 46, lines 209 - 216, and lines 223 – 241 for these changes. Supplementary methods have been added to discussion how we determined conservation of GMP modification sites.

- 5) Figure 1 has been modified after realizing that we mistakenly used a SARS-CoV nsp12 K73A mutant in our studies, instead of the SARS-CoV-2 nsp12 mutant. To correct this, we removed the SARS-CoV nsp12 mutant data and replaced it with the SARS-CoV-2 nsp12 mutants R116A and R116K. This residue is also strictly conserved in EAV nsp9 (as residue R124) and SARS-CoV (as residue R116) as described by Lehmann *et al.*, whose mutation abolished EAV nsp9 nucleotidylation activity and prevented both EAV and SAR-CoV from replicating in host cells. As such, in this study, mutation of SARS-CoV-2 nsp12 R116 abolished nucleotidylation activity, thereby implicating the NiRAN domain as responsible for nucleotidylation. While we were editing this figure, the data in the original Supplementary figure 2 was moved into Figure 1.

We apologize for this confusion. These changes are found in Figure 1 and lines 54 – 75.

Responses to Reviewers 1's Critiques:

General comment: “Although the study provides convincing evidence for arterivirus nsp9- and coronavirus nsp12-mediated GMPylation of other proteins, this reviewer feels that the biological relevance of the in vitro observations obtained for recombinant proteins remains unclear.”

Reply: We greatly share your desire to understand the exact biological relevance of the GMP modification. Broader studies with live virus unfortunately require advanced BSL facilities that are not currently accessible to us. Such studies would also require time beyond the acceptable review period for this manuscript. Given the interest in the SARS-CoV-2 nucleotidylase activity as briefly described below, we believe our data should formally be released to the public as soon as possible for the benefit of the scientific community.

The revised manuscript now includes expanded discussion section that specifically addresses the likely relevance and importance of our findings in the biology of the virus. In particular, while we were performing the requested review experiments, Yan *et al.* published a Cell article implicating the nsp12 NiRAN domain and its nucleotidylase activity in the process of RNA capping (Cell. 2020 Nov 14;S0092-8674(20)31533-6. doi: 10.1016/j.cell.2020.11.016). Formation of the RNA cap involves transfer of a GMP molecule to the 5' end of RNA. Traditionally, before the GMP is transferred to RNA, it forms an intermediate with a protein, such as what we have demonstrated here (reviewed in Nucleic Acids Res. 2016 Sep 19;44(16):7511-26. doi: 10.1093/nar/gkw551). The Yan *et al.* paper leaves many mechanistic steps to be resolved such as the identification of this protein intermediate and/or the methodology that can be used to identify this intermediate.

Please find this discussion in lines 223 – 241 in the revised manuscript.

In addition, we performed all experiments requested to address your critiques as outlined below.

- 1) **“As a minimum, mutant forms of (EAV and SARS-CoV-2) nsp7 and SARS-CoV nsp12 with substitutions of residues that are suggested to be GMPylated should be included in the study. Also, given some remaining doubts (page 6, 102f.) regarding GMPylation of the extra Gly-1 versus the natural Ser (Ser-2 in the recombinant protein) it would be desirable to produce and analyze an nsp7 protein with an authentic N-terminal Ser-1, along with mutant proteins in which the Ser proposed to carry an O-linked (rather than N-linked) GMP is replaced with Ala.”**

Reply: As suggested, we performed alanine point mutations to the GMPylation sites that were predicted from the mass spectrometry search software. We then assessed whether the GMP modification was still found on the peptide.

- a) For EAV nsp7 we found that mutation of K143, K156 and K172 to alanine eliminated the GMP modification on relevant peptide as assessed by mass spectrometry. This was also true for the EAV nsp9 GMPylation site at K380. However, the mutation of EAV nsp7 T3 to alanine did NOT result in the elimination of the modification. We also compared the radiolabeling of the EAV nsp7 and nsp9 mutant proteins to WT protein and found no difference. This was not surprising since we surmised there were multiple sites on these proteins thus elimination of a single site of many would not eliminate radiolabeling.

- b) For SARS-CoV-2, the gly-1 position is not found in the natural live virus and remains leftover as the penultimate position in the TEV cleavage site that was used to remove an N-terminal tag, as noted in the manuscript. It was suggested that the N-terminal glycine should be removed from the protein. Unfortunately, elimination of the glycine also abolished TEV cleavage for this specific protein.
- c) However, we continued with the SARS-CoV-2 nsp7 mutational analysis as suggested with the glycine in position 1.

A main concern was error in the prediction of the GMP modification site, particularly for the SARS-CoV-2 nsp7 modification site. We were also concerned that GMP-attachment site could shuffle during sample analysis from lys-3 to ser-2 because of previous studies discussed in the paper (see Anal Chem. 2007 Oct 1;79(19):7450-6. doi: 10.1021/ac0707838 , Anal Chem. 2008 Dec 15;80(24):9735-47. doi: 10.1021/ac801768s, J Mass Spectrom. 2009 Jun;44(6):861-78. doi: 10.1002/jms.1599, Anal Chem. 2019 Jan 2;91(1):126-141. doi: 10.1021/acs.analchem.8b04746, and Proteomics. 2013 Mar;13(6):945-54. doi: 10.1002/pmic.201200240)

Thus, we generated the following mutations: S2A, K3A, and S2AK3A. When K3A or S2AK3A mutations were introduced, we saw a 80-90% reduction in SARS-CoV-2 nsp7 radiolabeling with ³²P-GMP. However, mutation of S2A alone had no effect, indicating that the serine is NOT essential in nsp7 nucleotidylation. A similar reduction in labeling with the K3A mutations were observed by mass spectrometry. However, we still observed some residual nsp7 labeling on these mutant peptides by mass spectrometry. We suggest that the nearby N-terminus may serve as an alternate acceptor for GMP at a much-reduced efficiency; however, we acknowledge that we cannot provide validated explanation for how the residual amount of GMP labeling occurs. The signal for these GMP-modified, mutant peptides is much smaller, and the charge state is lower, which made it impossible to study the site assignment, as we did for the WT peptide.

These changes have been introduced into Table 1, the new Figure 4, lines 124 – 146 and lines 186 - 193. In addition, Supplementary Figures 10-13 have been provided to support this data. A statement has been added on lines 121 - 122 noting the reason why we could not remove the N-terminal glycine for the recombinant SARS-CoV-2 nsp7.

- 2) “ . . . **additional data should be provided to measure the transfer efficiency of the reaction** “

Reply: We now provide an estimate of total protein labeling using the radioactive signal from samples bands and a standard curve. Labeling percentages were estimated using the total protein concentration of the labeled proteins (also now provided as requested; see reply to critique #4 below) and curve fitting as now described in the methods. We performed this calculation for both the lower nucleotide concentration of 0.2 μM and at the higher concentration of nucleotide 200 μM. The higher concentration was used for mass spectrometry labeling experiments and is also close to physiological concentrations of nucleotide.

Not surprisingly, the labeling efficiency of the protein increased dramatically at higher nucleotide concentrations to 1 - 4.8%. At lower concentrations of nucleotide (0.2 μ M), which was below protein concentrations, labeling efficiency of the proteins was reduced to 0.006 – 0.03%, which corresponded to 0.06% to 0.3% of the total nucleotide. The increase in labeling efficiency is apparent because the radioactive band intensities are only ~10 fold lower at 1000x fold higher nucleotide concentrations (i.e. the radioactivity is diluted out by 1000 fold), representing ~100 fold increase in labeling efficiency.

This can be found in Fig. 5a and lines 150-153 of the revised manuscript. The experimental procedure to quantify the efficiency was added to the methods section (lines 294 – 298).

3) **“Also, the conservation in other arteri- and coronaviruses of residues identified to be GMPylated should be discussed.”**

Reply: We provided a discussion of the conservation of the confirmed GMP modified residues, which can be found in lines 209 – 216. Supplementary methods were added to discuss how we determined conservation.

4) **“Please provide information on the protein concentrations used in the nucleotidylation assay (page 18).”**

Reply: The protein concentrations have been added as requested. See lines 150 – 155 and lines 275 – 279 (methods) in the revised manuscript.

5) **“Given that Lehmann et al. reported that EAV nsp9 autonucleotidylation is more efficient with UTP compared to GTP, I suggest that experiments using UTP be performed.”**

Reply: We performed these experiments as suggested. We found UTP labeled both EAV and SARS-CoV-2 proteins. UTP appeared to label SARS-CoV-2 nsp7 and nsp8 more efficiently, while we found GTP yielded the most efficient labeling for EAV nsp7 and nsp9, contrasting to what Lehmann *et al.* found. We also performed a competition with cold ATP, CTP, UTP and GTP to outcompete GTP radiolabeling. Consistent with Lehman *et al.*, ATP and CTP minimally competed with ³²P-GTP labeling of the EAV proteins. Finally, because we observed strong UTP labeling of SARS-CoV-2, we continued our experiments and verified that a UMP covalent attachment was added to the same N-terminal SARS-CoV-2 nsp7 peptide via LC-MS/MS.

These data can be found in lines 147-171 and Figure 5.

6) **“Page 4, 63f. The paper by Chen et al (Cell 2020, 182(6):1560-1573.e13. doi: 10.1016/j.cell.2020.07.033. Epub 2020 Jul 28) should be cited in the context of the presumed nucleotidyl transferase active site.”**

Reply: Thank you for pointing an additional publication that is highly relevant to this work. We have added this as reference 19 found on lines 67 - 68 and also lines 203 – 204.

Responses to Reviewers 2's Critiques:

- 1) **“The authors have used nucleotides which concentration is higher than that for radio-labeling by near 3 orders of magnitude for in-vitro nucleotidylation reaction. It is not clear why nucleotidylation reactions were still performed under denatured conditions, 6.7 M Urea in 50 mM ammonium bicarbonate and 5mM DTT. This made the whole reaction condition greatly deviate from the cellular condition. I suggest to incubate high dosages under native conditions.”**

Reply: We apologize for the confusion, but the nucleotidylation reactions were in fact performed under native conditions (without Urea) as described in the methods section titled “Nucleotidylation Assays.” Then for subsequent LC/MS/MS processing, those completed reactions were denatured in the 6.7 M urea with 50 mM ammonium bicarbonate as described in the methods section titled “LC-MS/MS.” To help clarify this for the reader, we included statements in the methods section that indicates the reactions were performed under native conditions and were only denatured after the reaction was completed for LC-MS. Thus, the mass spectrometry experiments used 0.2 – 1 mM GTP under native conditions, exactly as you have suggested in order to approximate cellular conditions.

In response to your request, we performed radiolabeling using 200 μ M nucleotide and compared the labeling at the lower concentrations of 0.2 μ M. Labeling with both GTP and UTP was performed as requested by reviewer #1. We also estimated the percentage of protein labeling as reviewer #1 requested.

Radiolabeling was initially performed at \sim 0.2 μ M to replicate the finding of Lehmann *et al.*, who published similar findings with the EAV nsp9 protein. In addition, using a lower amount of total nucleotide typical results in high sensitivity, i.e. the radioactive signal is higher because the radionucleotide is a larger percentage of the total. This can be observed in the new Figure 5a. Of course, the percentage of protein that is labeled is decreased when using 0.2 μ M in part because this amount of total nucleotide is less than the actual protein amount 0.6 – 2.7 μ M.

Generally, using the higher, physiological concentrations of nucleotide yielded similar results, except that the protein labeling percentage was higher.

Changes in the manuscript can be found in Figure 5 and lines 147 – 155 in addition to statements added to the methods section at lines 275 – 276 and 300 – 301.

- 2) **“It is not clear why 15 mM DTT was used to neutralize unreacted iodoacetamide. Most protocols use DTT to break (reduce) disulfide linkages followed by alkylation rather than quenching extra iodoacetamide as they used. Explanations of their rationales are required.”**

Reply: The use of DTT to quench unreacted iodoacetamide is one common variation of in-solution digestion protocols that are used by enzyme manufacturers and proteomic centers.

For example, Peirce instructs users to quench the IAA in their instructions for MS grade trypsin (catalogue number 90057) solution digests (Please see the instructions at <https://www.thermofisher.com/order/catalog/product/90057?ICID=search-90057#/90057?ICID=search-90057>). The proteomic facilities at UC Davis and University of Washington also both provide protocols that has the user quench the IAA reaction with DTT (see the “in solution digestion protocol” at <http://proteomics.ucdavis.edu/protocols> and see page 2 of the following document: http://proteomicsresource.washington.edu/docs/protocols03/UWPR_Protocols_Protein_Digestion_Protocols.pdf).

The reason for this adaption is because IAA is known to overalkylate proteins. Treatment with IAA quickly results in the alkylation of the sulfhydryl sidechains of cysteine, which prevents disulfides from forming. This is the intended purpose of the IAA treatment. However, significant off-target modifications occur with the use of IAA that decrease the quality of LC/MS/MS data (Please see <https://doi.org/10.1021/ac0103423>, doi: 10.1007/978-1-4939-9232-4_7 and <http://dx.doi.org/10.3724/SP.J.1123.2013.04041> 0.3724/SP.J.1123.2013.04041 and the references therein). Thus, IAA is quenched in many protocols to prevent overalkylation, and importantly, the MS digestion enzymes function properly in the presence of DTT.

A statement was added to the methods, along with a relevant reference, in order to provide this explanation to readers. Please see lines 302 – 303.

Reviewers' comments:

Reviewer #1 (Remarks to the Author):

Conti et al. have addressed some of the points raised in a review of an earlier version of this manuscript. Specifically, the authors now provide evidence that, following substitution of 4 Lys residues in EAV nsp7 and EAV nsp9, the respective GMP-peptide adducts are no longer detected, confirming the predicted modification sites in most (but not all) cases. They also obtained evidence that SARS-CoV-2 nsp12 and EAV nsp9 mediate the transfer of UMP (in addition to GMP) to other proteins, albeit with different efficiencies. Also, attempts to assess the percentage of modification of the target proteins (1-4 % at high NTP concentrations) suggested that a rather small proportion is modified by the cognate NiRAN domain.

Although some additional evidence is now provided to support NiRAN-mediated guanylation/uridylation of coronavirus nsp7 (and nsp8) and EAV nsp7 at multiple sites, I have concerns regarding the biological significance of these findings. In the absence of additional control proteins, it cannot be excluded that the observed protein nucleotidylation represents a nonspecific (background) activity (artifact). This assessment is largely based on the following points:

1. The extent of nucleotidylation appears to be low for both substrate proteins (unfortunately, limited detail is provided on the experimental conditions used, complicating an independent assessment).
2. Nucleotidylation occurs at multiple Lys residues with no specific sequence context and at residues that do not appear to be conserved among related viruses.
3. Structural studies reported for coronavirus nsp7/8/12 complexes show that nsp7 and 8 form stable complexes with nsp12, and do not interact with surfaces close to the NiRAN active site. Can this be reconciled with the data of this study?
4. The biological relevance of the modifications has not been determined in the context of viral replication (to my knowledge, EAV work would not require BSL-3 containment conditions and would thus be feasible to do for most laboratories).

Sequence alignments with orthologs from other corona- and arteriviruses, respectively, are not shown for the "target" proteins and the respective information provided in Supplementary methods is of poor quality. See, for example, the statement: "The polyprotein encodes the open reading frame that is cleaved into the nsp proteins". The extent of conservation of residues confirmed to be nucleotidylated is not appropriately addressed and discussed. It is not even mentioned that three of the modified Lys residues are located in nsp7beta.

Overall, the data support the idea that arterivirus and coronavirus polymerase proteins (through their NiRAN domains) have protein nucleotidylation activities. The study however fails to provide convincing evidence that the coronavirus nsp7 and nsp8 and the arterivirus nsp7(beta) proteins are relevant substrates for these activities. Protein (auto)nucleotidylation activity has been shown previously for the EAV nsp9 (Lehmann et al., 2015), further limiting the novelty and significance of this study.

Text and figures (see, for example Figs. 2 and 5) need to be thoroughly revised to remove typos and use consistent protein and virus names.

Reviewer #2 (Remarks to the Author):

The authors have performed extra experiments and answered most of my questions. For the concentration issue, although the authors had mis-understood my concerns and added radioisotope experiments with higher concentrations rather than using lower concentrations with

MS experiment which was my original thought, I feel OK with the data since the authors have added mutant experiments which did show the K-nucleotidylation was suppressed by mutation.

Changes made in manuscript revision COMMSCHEM-20-0326B by Conti *et al.* in response to reviewer comments:

- 1) **Reviewer 1 commented “Nucleotidylation occurs at multiple Lys residues with no specific sequence context and at residues that do not appear to be conserved among related viruses.”**

... “Sequence alignments with orthologs from other corona- and arteriviruses, respectively, are not shown for the “target” proteins and the respective information provided in Supplementary methods is of poor quality. See, for example, the statement: “The polyprotein encodes the open reading frame that is cleaved into the nsp proteins”. The extent of conservation of residues confirmed to be nucleotidylated is not appropriately addressed and discussed. It is not even mentioned that three of the modified Lys residues are located in nsp7beta.”

In our previous version of the manuscript (COMMSCHEM-20-0326T), we stated on lines 209-210 and 214-216 that there is conservation of the modified EAV nsp9 and SARS-CoV-2 nsp7 GMP modified residues. We also briefly discussed and attempted to convey that three of the EAV nsp7 GMP-modified residues are in nsp7 β in lines 212-214. Thus, we attempted to clarify and expand upon our statements.

We now have added the requested alignments as Supplementary Figure 14-17 along with two additional pages of methods in the Supplementary information to provide in depth details on how conservation was assessed. We tried to further clarify the scientific meaning of “polyprotein” for readers and provided a reference that explains the nitty-gritty details for how viruses make the non-structural proteins from one polyprotein peptide via post-translation peptidase cleavage. We also expanded and rearranged our discussion on the conservation in the main text, including explicitly stating that three of the EAV GMP-modification sites are in the nsp7b portion of the protein as requested.

We are however cautious to draw firm conclusions from conservation or lack of conservation because only experimentation can reveal their significance.

See lines 214 – 233 in the revised manuscript, the supplementary methods section (pages 2-4) in the supplementary information, and supplementary figures 14-17.

- 2) **Reviewer 1 commented that “Structural studies reported for coronavirus nsp7/8/12 complexes show that nsp7 and 8 form stable complexes with nsp12, and do not interact with surfaces close to the NiRAN active site. Can this be reconciled with the data of this study?”**

Reviewer 1 stated this as a reason for why GMP-modification of SARS-CoV-2 nsp7 and nsp8 by nsp12 could be artifact.

To respond, we now expand our discussion to more explicitly state how SAR-CoV-2 nsp7 and nsp8 could be modified with GMP via nsp12. As the reviewer indicated, we originally pointed that the binding sites of nsp7 and nsp8 on nsp12 are distant from the nsp12 NiRAN domain (per

numerous structural studies in the literature). We now explicitly extrapolate for the reader on how nsp7 and nsp8 are likely modified by nsp12 in this study.

We briefly explain that in solution, protein complexes are in constant dynamic equilibrium with their monomer states and are assembled from their monomeric subunits. Thus, nsp7 and nsp8 would likely have to be modified by nsp12 before any association with nsp12.

See lines 209-212 in the revised manuscript.

3) Reviewer 1 commented that “The extent of nucleotidylation appears to be low for both substrate proteins (unfortunately, limited detail is provided on the experimental conditions used, complicating an independent assessment).”

Reviewer 1 stated this as a reason for why GMP-modification of the substrates documented in this study could be artifactual.

To our knowledge, we have provided the exact steps of these labeling reactions and how we quantitated in the methods section such that anyone can understand exactly how we determined the extent of nucleotidylation. Unfortunately, without more detail, we are not sure what the reviewer is seeking to understand.

To our knowledge, no one extrapolates biological, in vivo meaning from absolute enzymatic efficiencies measured in vitro. In fact, studies routinely do not provide any details on absolute efficiency, and do not compare them to in vivo activity. This is because it is difficult, if not impossible, to model how factors such as protein stability, lack of additional cofactors, buffer conditions, etc alter the in vitro activities. In addition, exact efficiency of in vivo enzymatic activity is also difficult to measure and know due to the low relative abundance of the product and competing reactions that remove the product. It is the relative efficiencies that are used routinely in vitro, such as comparing the efficiency of a WT and mutant protein as we have done in this study in order to assess whether we determined the correct modification sites.

We now state for the readers that it is difficult to draw conclusions from the efficiency of in vitro biochemical reaction on how it relates to the in vivo activity due to a number of factors such as protein stability, lack of additional cofactors, and buffer conditions. We provide a reference that discusses these difficulties in detail in relation to the RNA polymerase activity of the EAV and SARS-CoV enzymes.

See lines 152-156 in the revised manuscript.

4) Reviewer 1 commented that “In the absence of additional control proteins, it cannot be excluded that the observed protein nucleotidylation represents a nonspecific (background) activity (artifact).”

We now explicitly point out that nsp12 is in fact an additional internal control protein that is not modified with GMP, intermolecularly (Figure 1b). This is perhaps an even more relevant control

than BSA. Given that fact that reviewer 1 did not recognize other information that we included in our previous revised manuscript, we wanted to explicitly state this additional protein control.

We also now caution the reader explicitly that in vivo studies are needed to understand the biological role of the described nucleotidylase activity and to confirm these substrates.

See lines 73-75 and 257-260 in the revised manuscript.

5) Reviewer 1 commented that “Text and figures (see, for example Figs. 2 and 5) need to be thoroughly revised to remove typos and use consistent protein and virus names.”

Throughout the manuscript, a handful of typos and inconsistent capitalizations were found and fixed. These corrections are not highlighted in the revised manuscript.